# PAINT BY INPAINT: LEARNING TO ADD IMAGE OBJECTS BY REMOVING THEM FIRST

## ABSTRACT

Image editing has advanced significantly with the introduction of text-conditioned diffusion models. Despite this progress, seamlessly adding objects to images based on textual instructions without requiring user-provided input masks remains a challenge. We address this by leveraging the insight that removing objects (`Inpaint`) is significantly simpler than its inverse process of adding them (`Paint`), attributed to the utilization of segmentation mask datasets alongside inpainting models that inpaint within these masks. Capitalizing on this realization, by implementing an automated and extensive pipeline, we curate a filtered large-scale image dataset containing pairs of images and their corresponding object-removed versions. Using these pairs, we train a diffusion model to inverse the inpainting process, effectively adding objects into images. Unlike other editing datasets, ours features natural target images instead of synthetic ones; moreover, it maintains consistency between source and target by construction. Additionally, we utilize a large Vision-Language Model to provide detailed descriptions of the removed objects and a Large Language Model to convert these descriptions into diverse, natural-language instructions. Our quantitative and qualitative results show that the trained model surpasses existing models in both object addition and general editing tasks. To propel future research, we will release the dataset alongside the trained models.

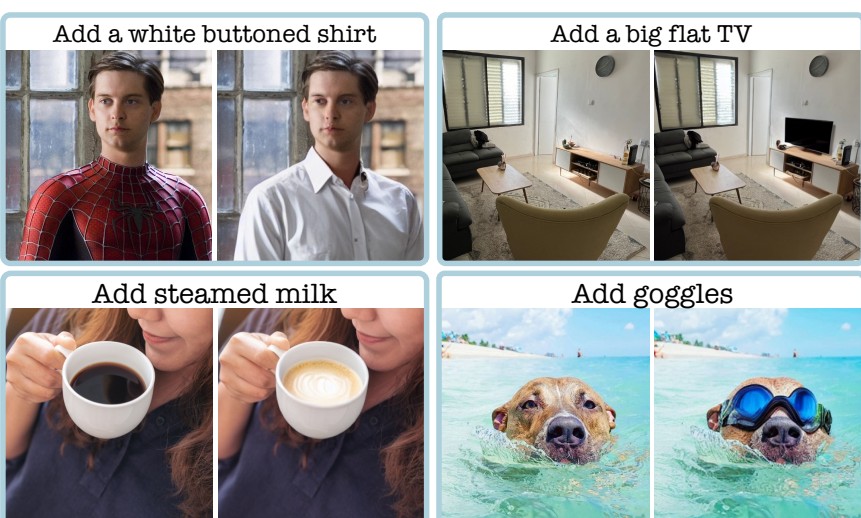

Figure 1: **Visual Results of the Models Trained with the Proposed Dataset.**

## 1 INTRODUCTION

Image editing plays a central role in the computer vision and graphics communities, with diverse applications spanning various domains. The task is inherently challenging as each image offers infinite editing possibilities, each with countless potential outcomes. A particularly intricate editing

task is seamlessly adding objects to images, which requires not only realistic visuals but also a nuanced understanding of the global image context, including parameters such as location, scale, and style. While many solutions require the user to provide a mask for the target object (Li et al., 2023b; Xie et al., 2023; Rombach et al., 2022; Wang et al., 2023a), recent advancements have capitalized on the success of text-conditioned diffusion models to enable a mask-free approach (Brooks et al., 2023; Zhang et al., 2023). Such solutions offer a more convenient and realistic setting; yet, they still encounter challenges, as demonstrated in Figure 3.

The leading method for such editing, InstructPix2Pix (IP2P) (Brooks et al., 2023), synthesizes a dataset containing triplets of source and target images alongside an editing instruction as guidance. Under this guidance, a model is trained to transform source images into target ones. While demonstrating some success, the model's effectiveness is bounded by the quality of the synthesized training data. We address this limitation by introducing an alternative automatic method for creating a large-scale, high-quality dataset targeted for image object addition. Our approach is grounded in the observation that adding objects (`paint`) is essentially the inverse of removing them (`inpaint`). Namely, by using pairs of images—ones containing objects and others with objects removed—an object addition dataset can be established. In practice, we create the dataset by leveraging abundant images and object masks available in segmentation datasets (Kuznetsova et al., 2020b; Lin et al., 2014; Gupta et al., 2019) alongside a high-end inpainting model (Rombach et al., 2022). The outputs are then used in a reverse manner, with the original images as editing targets and the inpainted ones as sources. This reversed approach is essential because directly adding objects with an inpainting model requires object segmentations not present in the images. **Our approach offers two key advantages over IP2P:** (i) While IP2P relies on synthetic source and target images, our targets are real natural images, with source images also being natural outside the typically small edited regions. (ii) Despite employing techniques such as prompt-to-prompt (Hertz et al., 2022) and Directional CLIP-based filtering (Gal et al., 2021) to address source-target consistency issues, IP2P often fails to achieve this. In contrast, our approach inherently maintains consistency by construction.

Mask-based inpainting models have recently shown great success in filling image masks naturally and coherently (Rombach et al., 2022). However, since these models were not trained specifically for object removal, their use for this purpose is not guaranteed to be artifact-free, potentially leaving remnants of the original object, unintentionally creating new objects, or causing other distortions. Given that the outputs of inpainting serve as training data, these artifacts could potentially impair the performance of the resulting models. To counteract these issues, we propose a comprehensive pipeline of varied filtering and refinement techniques. Additionally, we complement the source and target image pairs with natural language editing instructions by harnessing advancements in multimodal learning (Li et al., 2023a; Dai et al., 2023; Liu et al., 2023; Bai et al., 2023; Ganz et al., 2023; 2024; Rotstein et al., 2023). By employing a Large Vision-Language Model (VLM) (Wang et al., 2024b), we generate elaborated captions for the target objects. Next, we utilize a Large Language Model (LLM) (Jiang et al., 2023) to cast these descriptions to natural language instructions for object addition. To further enhance our dataset, we incorporate human-annotated object reference datasets (Kazemzadeh et al., 2014; Mao et al., 2016) and convert them into adding instructions. Overall, we combine these sources to form an instruction-based object addition dataset, named PIPE (**P**aint by **Inp**aint **E**diting). Unprecedented in size, our dataset features approximately 1 million image pairs, spans over $1400$ different classes, and includes thousands of unique attributes.

Utilizing PIPE, we train a diffusion model to follow object addition instructions, setting a new standard for adding realistic image objects, as demonstrated in Figure 1, and as validated across extensive experiments on multiple benchmarks. Besides quantitative results, we conduct a human evaluation survey comparing our model to top-performing models, showcasing its improved capabilities. Furthermore, we demonstrate that PIPE can extend beyond mere object addition; by integrating it with additional editing datasets, we show it significantly improves overall editing results.

**Our contributions include:**

- Introduction of the *Paint by Inpaint* framework for image editing.

- Construction of PIPE, a large-scale, high-quality, mask-free, textual instruction-guided object addition image dataset.

- Demonstration of a diffusion-based model trained with PIPE, achieving state-of-the-art performance in adding objects to images and enhancing general editing performance.

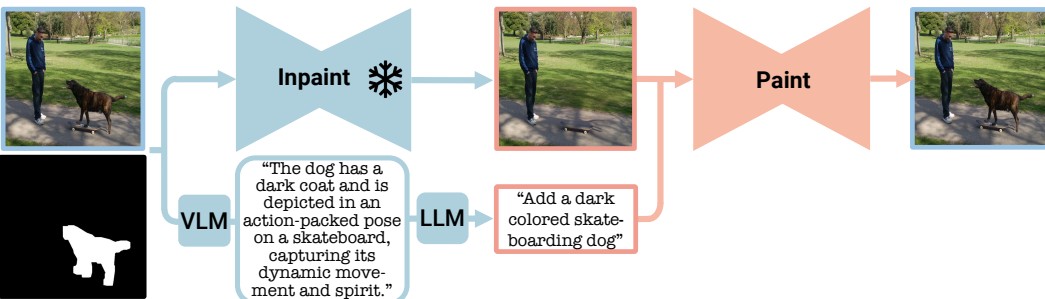

Figure 2: **Paint by Inpaint Framework.** Illustration of our two-phase approach: (1) Building PIPE dataset (blue), which involves: (i) Removing the object utilizing a frozen inpainting model and the object mask. (ii) Generating addition instructions, demonstrated through the VLM-LLM-based procedure, where a VLM extracts visual object details and an LLM formulates them into instructions. (2) Training an editing model (orange), PIPE is employed to train a model to reverse the inpainting process, thereby adding objects to images.

## 2 RELATED EFFORTS

### 2.1 IMAGE EDITING

Image editing has long been explored in computer graphics and vision (Oh et al., 2001; Pérez et al., 2023). The field has seen substantial advances with the emergence of diffusion-based image synthesis models (Song et al., 2020; Ho et al., 2020), especially with their text-conditioned variants (Ramesh et al., 2022; Rombach et al., 2022; Saharia et al., 2022b; Nichol et al., 2021). The application of such models can be broadly categorized into two distinct approaches – mask-based and mask-free.

**Mask-Based Editing.** Such approaches formulate image editing as an inpainting task, using a mask to outline the target edit region. Early diffusion-based techniques utilized pretrained models for inpainting (Song et al., 2020; Avrahami et al., 2022; Yu et al., 2023; Meng et al., 2021), while more recent approaches fine-tune the models specifically for this task (Nichol et al., 2021; Saharia et al., 2022a; Rombach et al., 2022). Inpainting models benefit from the possibility of training on large-scale image datasets, as they can be trained with any image paired with a random mask. Various attempts have been made to advance this methodology in different directions (Wang et al., 2023a; Li et al., 2023b; Xie et al., 2023), but despite this progress, relying on a user-provided mask makes this setting less preferable in real-world applications.

**Mask-Free Editing.** This paradigm allows image editing using text and natural language as an intuitive interactive tool without the need for additional masks. Kawar *et al.* (Kawar et al., 2023) optimize a model to align its output with a target embedding text. Bar Tal *et al.* (Bar-Tal et al., 2022) introduce a model that merges an edit layer with the original image. IP2P turns mask-free image editing into a supervised task by generating an instruction-based dataset using Prompt-to-Prompt (Hertz et al., 2022) and an LLM (Brooks et al., 2023). The Prompt-to-Prompt technique adjusts cross-attention layers in diffusion models, aligning attention maps between source and target prompts. These mask-free techniques are distinguished by their ability to perform global edits such as style transfer. However, they exhibit limitations in local edits, specifically in maintaining consistency outside the desired edit region. IP2P seeks to address this by utilizing Directional CLIP loss (Gal et al., 2021) for dataset filtering. Nevertheless, it mitigates the limitation, but only to some extent. In contrast, our dataset ensures consistency by strictly limiting changes to the intended edit regions only.

**Instructions-Based Editing.** A few studies have introduced textual instructions for intuitive, mask-free image editing without complex prompts (El-Nouby et al., 2019; Zhang et al., 2021). IP2P facilitates this by leveraging GPT-3 (Brown et al., 2020) to create editing instructions from input image captions. Following the advancements in instruction-following capabilities of LLMs (Ouyang et al., 2022; Ziegler et al., 2019), Zhang *et al.* devise a reward function reflecting user preferences on edited images (Zhang et al., 2023). Our approach takes a different course; it enriches the class-based instructions constructed from the segmentation datasets by employing a VLM (Wang et al., 2023b) to comprehensively describe the target object, and an LLM (Jiang et al., 2023) to transform

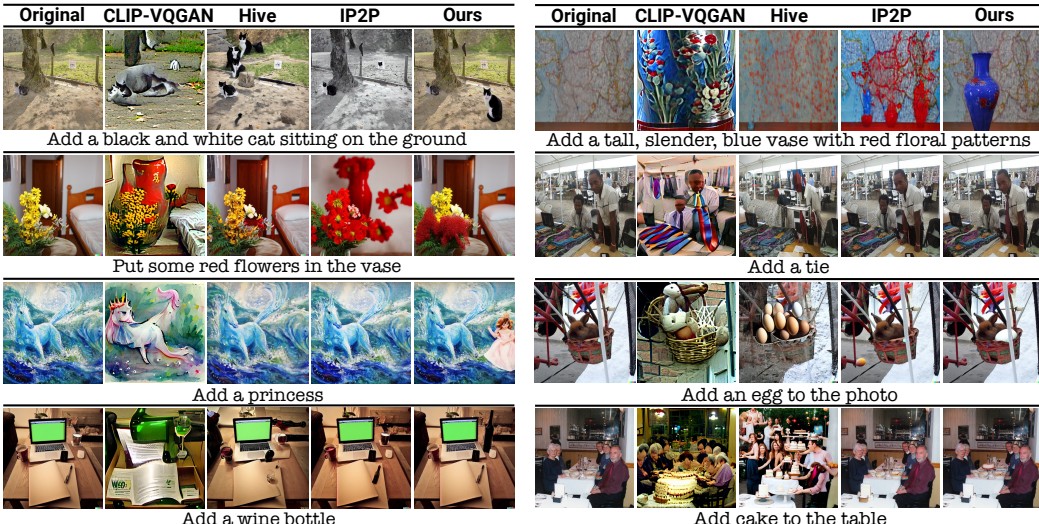

| Original | CLIP-VQGAN | Hive | IP2P | Ours |
| --- | --- | --- | --- | --- |

Add a black and white cat sitting on the ground

Put some red flowers in the vase

Add a princess

Add a wine bottle

Add a tall, slender, blue vase with red floral patterns

Add a tie

Add an egg to the photo

Add cake to the table

Figure 3: **Visual Comparison.** Comparison of our model with leading editing models across different benchmarks, demonstrating superior fidelity to instructions and precise object addition in terms of style, scale, and position, while maintaining higher consistency with original images.

the VLM outputs into coherent editing instructions. Our dataset is further enhanced by integrating object reference datasets (Kazemzadeh et al., 2014; Mao et al., 2016), which are converted into compositional, rich, and detailed instructions.

## 2.2 IMAGE EDITING DATASETS

Early editing approaches (Xu et al., 2018; Zhang et al., 2017) used datasets with specific classes without direct correspondence between source and target images (Lin et al., 2014; Wah et al., 2011; Nilsback & Zisserman, 2008). Building datasets of natural images and their natural edited versions in the mask-free setting is infeasible, as it requires two identical images differing solely in the edited region. Thus, previous works propose synthetic alternatives, with the previously discussed IP2P's dataset being one of the most prominent ones. MagicBrush (Zhang et al., 2024) recently introduced a partially synthetic dataset, which was manually created using DALL-E2 (Ramesh et al., 2022). While offering more accuracy and consistency, its manual annotation and monitoring limit its scalability. Inst-Inpaint (Yildirim et al., 2023) leverages segmentation and inpainting models to develop a dataset focused on object removal, designed to eliminate the segmentation step. We introduce a high-quality image editing dataset that exceeds the scale of any currently available ones. Furthermore, our approach, uniquely leverages real images as the edit targets, distinguishing it from prior datasets consisting of synthetic data.

## 2.3 OBJECT FOCUSED EDITING

Processing specific objects through diffusion models has gained significant attention in recent research. For instance, various methodologies have been developed to generate images of particular subjects (Ruiz et al., 2023; Gal et al., 2022a; Chen et al., 2024). Within the editing domain, Wang *et al.*(Wang et al., 2023a) concentrate on mask-based object editing, training their model for inpainting within existing object boundaries, while Patashnik *et al.*(Patashnik et al., 2023) introduce a technique for producing diverse variations of such objects. Similar to our work, SmartBrush (Xie et al., 2023) aims to add objects to images. However, unlike our methodology, it requires an input mask from the user. Instruction-based methods like IP2P and MagicBrush highlight their capability to insert image objects, **allocating a considerable portion of their dataset for this purpose,** for example, 39% of the MagicBrush dataset is dedicated to this task.

## 3 PIPE DATASET

As outlined in Section 2, leading mask-free, instruction-following image editing models are trained on datasets that are either small-scale or synthetic and inconsistent. To enhance the efficacy of these models, we propose a systematic method to create a dataset that addresses these limitations. The

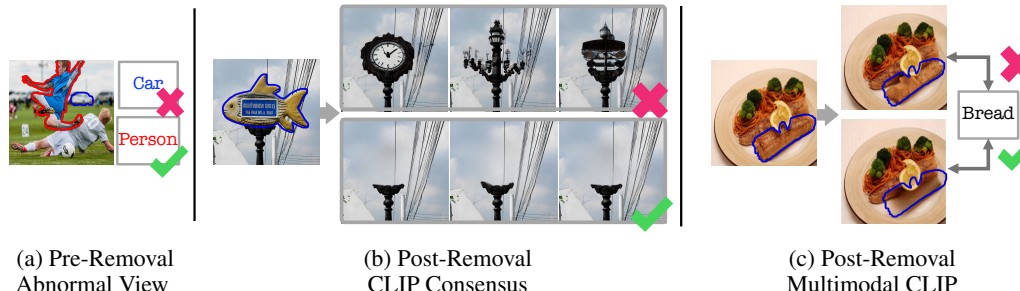

(a) Pre-Removal
Abnormal View

(b) Post-Removal
CLIP Consensus

(c) Post-Removal
Multimodal CLIP

Figure 4: **Dataset Filtering Stages.** In constructing PIPE, several filtering stages address inpainting drawbacks. Initially, a pre-removal filter targets abnormal object views due to blur and low quality. Subsequently, a post-removal inconsistency filter identifies a lack of CLIP consensus among three inpainting outputs, indicating substantial variance and potential object regeneration. Finally, a post-removal multimodal CLIP filtering ensures low semantic similarity with the original object name.

devised dataset, dubbed PIPE (**P**aint by **InP**aint **E**dit), comprises approximately 1 million image pairs accompanied by diverse object addition instructions. Our methodology, illustrated in blue in Figure 2, unfolds in a two-stage procedure. First, drawing on the insight that object removal is more straightforward than object addition, we create pairs of source and target images—without and with objects. Subsequently, we generate a natural language object addition instruction for each pair using various techniques. In the following section, we describe the proposed pipeline in detail.

### 3.1 GENERATING SOURCE-TARGET IMAGE PAIRS

In the initial stage of creating PIPE, we leverage extensive image segmentation datasets. Specifically, we utilize COCO (Lin et al., 2014) and Open Images (Kuznetsova et al., 2020a), enriched with segmentation mask annotations from LVIS (Gupta et al., 2019). Unifying these datasets results in $889,230$ unique images with over $1,400$ object classes. We use this diverse corpus for object removal using a Stable Diffusion (SD) (Rombach et al., 2022) based inpainting model[1]. This configuration is the underlying reason why constructing PIPE via removal is more straightforward than via addition. However, since the inpainting model was not trained specifically for object removal, it can yield suboptimal outcomes, e.g., leaving original object traces or generating new objects. To address this, we implement a pipeline of pre-removal and post-removal steps.

**Pre-Removal.** This step filters object segmentation masks, retaining only candidates suitable for the subsequent object-adding. First, we exclude masks according to their size (too large or too small) and location (near image borders). Next, we use CLIP (Radford et al., 2021) to calculate the semantic similarity between segmented objects and their class names, using low values to filter out abnormal object views (*e.g.*, blurred objects) and non-informative partial views (*e.g.*, occluded objects). In Figure 4a, we provide an example of a car being filtered due to its small size and blur, while a person without these characteristics is not (see fig. S9 for more examples). To ensure the mask fully covers the object, we apply morphological dilation, a crucial step since any unmasked object parts can lead the inpainting model to regenerate it (Pobitzer et al., 2024).

**Object Removal.** Given the dilated masks, we remove the objects using the SD inpainting model. Unlike conventional inpainting objectives, which aim at general image completion, our focus centers on object removal. To this end, we guide the model with positive and negative prompts designed to replace objects with non-objects (*e.g.*, background). The positive prompt is set to "`a photo of a background, a photo of an empty place`", while the negative prompt is defined as "`an object, a <class>`", where `<class>` denotes the object class name. During the inpainting process, we utilize 10 diffusion steps and generate 3 distinct outputs per input.

**Post-Removal.** The last part of our removal pipeline involves employing a multi-step process aimed at filtering and refining the inpainting outputs:

- Removal Verification: For each source image and its three inpainted outputs, we introduce two mechanisms to assess removal effectiveness. First, we measure the semantic diversity of the three inpainted candidates' regions by calculating the standard deviation of their CLIP embed-

---

[1]`https://huggingface.co/runwayml/stable-diffusion-inpainting`

| Source | Target | Source | Target | Source | Target | Source | Target |
|--------|--------|--------|--------|--------|--------|--------|--------|

| Add a bus | Add a light-colored plastic frisbee | Add a black round hat with a flat top | Add a bird closest to camera |

Figure 5: **PIPE dataset Examples.** Samples from PIPE using different instruction generation techniques: class name-based (left), VLM-LLM based (center), and reference-based (right).

dings, a metric we refer to as the CLIP consensus. Intuitively, high diversity (no consensus) suggests failed object removal, leaving varied non-background object elements, as shown in the upper row of Figure 4b. Conversely, lower variability (consensus) points to a consistent removal, increasing the likelihood of an appropriate background, as demonstrated in the bottom row of the figure. Next, we calculate the CLIP similarity between the inpainted region of each candidate and the class name of the removed object (e.g., <bread>). This procedure, referred to as multimodal CLIP filtering, is illustrated in Figure 4c. Introducing CLIP consensus and multimodal CLIP filtering mechanisms enhances the robustness of the object removal process. If multiple candidates pass all filtering stages, the one with the lowest multimodal CLIP score is selected. Prior to choosing the CLIP Consensus and Multimodal CLIP filters thresholds, we manually annotated 500 inpainted images, classifying them as successful or failed removals. We tested the filters across varying thresholds and plotted the percentage of successful inpainted images against the percentage of filtered images. As shown in fig. S11 and fig. S12, as the filters become more aggressive (lower thresholds), the proportion of successful inpainted images increases for both strategies. This implies that both filtering approaches effectively achieve their aim of filtering out unsuccessful inpainting outputs. We selected thresholds where the slope of successful inpainting begins to plateau, minimizing the loss of images while maximizing quality.

- Consistency Enforcement: We aim to produce image targets that are consistent with the source ones. By conducting $\alpha$-blending between the source and inpainted image using the object mask, we limit differences to the mask area while ensuring a smooth, natural transition between regions (see example in fig. S10).

- Importance Filtering: In the final removal pipeline step, we filter out instances where the removed object has marginal semantic importance, as such edits are unlikely to be user-requested. We use a CLIP image encoder to assess the similarity between source and target images—not limited to the object region—filtering cases exceeding a manually set threshold.

### 3.2 GENERATING OBJECT ADDITION INSTRUCTIONS

The PIPE dataset is designed to include triplets of source and target images, along with corresponding editing instructions in natural language. However, the process outlined in Section 3.1 only produces pairs of images and the raw class name of the object of interest. To address this gap, we introduce three different strategies for enhancing our dataset with instructions:

**Class name-based instructions.** We augment raw object classes into object addition instructions using the format "add a <class>", leading to simple and concise instructions.

**VLM-LLM based instructions.** We propose an automatic procedure designed to produce more varied and comprehensive instructions than those based on class names. Leveraging recent VLM and LLM advances, we craft instructions using a two-stage process, as illustrated in Figure 2. In the first stage, we mask out non-object regions and insert the devised image into a VLM, namely CogVLM[2] (Wang et al., 2024b), prompting it to generate a detailed object caption that includes visual object details and fine-grained attributes. In the second stage, the caption is reformatted into an instruction using the in-context learning (ICL) capabilities of the LLM. Specifically, we utilize Mistral-7B[3] (Jiang et al., 2023) with 5 ICL examples of the required outputs, prompting it to generate instructions of varying lengths and complexity. This two-stage process, designed to mitigate hallucinations frequently encountered with VLMs (Liu et al., 2024), has been empirically validated

---

[2]https://huggingface.co/THUDM/cogvlm-chat-hf
[3]https://huggingface.co/mistralai/Mistral-7B-Instruct-v0.2

Table 1: **Datasets Comparison.** Review of PIPE with others editing datasets. ✓ signifies fulfillment, ✗ indicates non-fulfillment, and ✗ denotes partial fulfillment, where images are real outside inpainted areas. "–" means no such images available. "General Classes" indicates dataset class diversity.

| Dataset | Real Source Images | Real Target Images | General Classes | # Images | # Edits |
|---|---|---|---|---|---|
| Oxford-Flower Nilsback & Zisserman (2008) | ✓ | ✓ | ✗ | 8,189 | 8,189 |
| CUB-Bird Wah et al. (2011) | ✓ | ✓ | ✗ | 11,788 | 11,788 |
| EditBench Wang et al. (2023a) | ✗ | – | ✓ | 240 | 960 |
| InstructPix2Pix Brooks et al. (2023) | ✗ | ✗ | ✓ | 313,010 | 313,010 |
| MagicBrush Zhang et al. (2024) | ✓ | ✗ | ✓ | 10,388 | 10,388 |
| PIPE | ✗ | ✓ | ✓ | 889,230 | 1,879,919 |

as effective and is inspired by research demonstrating that breaking down tasks into specific model roles enhances LLMs performance (Wang et al., 2024a). Further details of this procedure are provided in the supplementary materials.

**Manual Reference-based Instructions.** To enrich our dataset with additional nuanced, compositional object details, we utilize three object reference datasets: RefCOCO, Ref-COCO+ (Kazemzadeh et al., 2014), and RefCOCOg (Mao et al., 2016). We transform the references into instructions using the template: "add a <object reference>", where "<object reference>" is replaced with the dataset's object description.

Incorporating these diverse approaches produces $1,879,919$ different realistic object addition instructions, encompassing both concise and detailed editing scenarios. Examples from PIPE using these diverse approaches are presented in Figure 5 and the appendix. In Table 1, PIPE is compared with other image editing datasets. It sets a new benchmark in image and editing instruction count by a significant margin. Notably, it is the only dataset offering real target images and class diversity.

## 4 Model Training

We detail the methodology used to train an image editing model using the proposed dataset, as illustrated in orange in Figure 2. We leverage the SD 1.5 model (Rombach et al., 2022) for both its architecture and initial weights. This text-conditioned diffusion model incorporates a pre-trained variational autoencoder and a U-Net (Ronneberger et al., 2015), which is responsible for the diffusion denoising within the latent space of the former. We denote the model parameters as $\theta$, the noisy latent variable at timestep $t$ as $z_t$, and the corresponding score estimate as $e_\theta$. Similar to SD, our editing process is conditioned on a textual instruction encoding $c_T$ through cross-attention which integrates text encodings with visual representations. We employ classifier-free guidance (CFG) (Ho & Salimans, 2022) to enhance alignment between the output image and the instruction encoding $c_T$. Contrary to SD, which generates a completely new image, our method involves editing an existing one. Thus, similarly to IP2P, we condition the diffusion process not only on $c_T$ but also on the input image, denoted as $c_I$. Liu et al. (Liu et al., 2022) demonstrated that a diffusion model can be conditioned on multiple targets, adapting CFG accordingly. Using CFG necessitates modeling both conditional and unconditional scores. To facilitate this, during training we set $c_T = \varnothing$ with probability $p = 0.05$ (no text conditioning), $c_I = \varnothing$ with $p = 0.05$ (no image conditioning), and $c_I = \varnothing, c_T = \varnothing$ with $p = 0.05$ (no conditioning). During inference, using CFG, we compute the following score estimate considering both the instruction and the source image,

$$
\begin{aligned}
\tilde{e}_\theta(z_t, c_I, c_T) = \; & e_\theta(z_t, \varnothing, \varnothing) \\
& + s_T \cdot (e_\theta(z_t, c_I, \varnothing) - e_\theta(z_t, \varnothing, \varnothing)) \\
& + s_I \cdot (e_\theta(z_t, c_I, c_T) - e_\theta(z_t, c_I, \varnothing)),
\end{aligned}
\tag{1}
$$

where $s_T$ and $s_I$ represent the CFG scales for the textual instruction and the source image, respectively. Further implementation details and hyperparameters are provided in the appendix.

## 5 Experiments

Image editing can yield countless different valid outcomes, making its evaluation a significant challenge. To address this, we perform a diverse array of experiments. Given that PIPE is primarily

designed for object addition, we initially focus our experiments on this task before extending its application to general editing (in Section 6). We quantitatively and qualitatively compare our model with top-performing methods, complemented by an in-depth detailed human evaluation survey. Additionally, in the appendix, we include an ablation study of the VLM-LLM pipeline.

## 5.1 Experimental Settings

We consider three benchmarks to evaluate our model's capabilities in object addition – (i) PIPE test set: 750 images from the COCO validation split, generated using the pipeline outlined in Section 3. (ii) OPA (Liu et al., 2021): An object placement assessment dataset that includes source and target images, along with objects to be added. (iii) MagicBrush (Zhang et al., 2024): A partially synthetic image editing benchmark comprising training and testing sets. To evaluate object addition, we automatically filter the dataset for this task (details in the appendix), resulting in a 144 edits subset.

## 5.2 Quantitative Evaluation

We compare our model with leading image editing models, including Hive (Zhang et al., 2023), IP2P (Brooks et al., 2023), VQGAN-CLIP (Crowson et al., 2022), SDEdit (Meng et al., 2021), Null-Text-Inversion (Mokady et al., 2023), Pix2PixZero (Parmar et al., 2023) and Edit-Freindly DDPM (Huberman-Spiegelglas et al., 2024). For evaluating objects additions, we use the standardized metrics from MagicBrush (Zhang et al., 2024). These metrics compare edited outcomes to ground-truth targets using both model-free ($L_1$ and $L_2$ distances) and model-based (CLIP (Radford et al., 2021) and DINO (Caron et al., 2021) embedding cosine distances) measures. Model-free metrics penalize global changes affecting non-object regions, while model-based approaches evaluate overall semantic similarity. When the edited target caption is available, we use CLIP-T (Ruiz et al., 2023) to measure its alignment with the edited image. To complement our evaluation, we adopt the recently proposed Conditional Maximum Mean Discrepancy (CMMD) metric (Jayasumana et al., 2024). Like the popular Fréchet Inception Distance (FID) (Heusel et al., 2017), this metric measures the distributional distance between groups of images. However, unlike FID, CMMD uses CLIP embeddings and works effectively with a reduced number of samples, enabling us to measure distribution distances for small datasets like MagicBrush. To further demonstrate the superiority of our model, we adopt a measure utilized by (Brooks et al., 2023). This measure, using changing image guidance scales ($s_I$), plots a graph of two metrics of the edited outcome, both independent of a ground-truth target image: (i) CLIP similarity with the input image. (ii) Directional CLIP similarity (Gal et al., 2022b), which evaluates changes between source-target image embeddings and source-target text caption embeddings. This plot presents a trade-off between preserving the original content and achieving the desired edits.

**PIPE Test Results**. We evaluate our model against instruction-following models, Hive and IP2P, using the PIPE held-out test set and report the results in Table 3. Our model significantly surpasses the baselines in $L_1$ and $L_2$ metrics, confirming its high consistency, and exhibits a higher level of semantic resemblance to the target ground truth image, as reflected in the CLIP-I and DINO scores.

**OPA Results**. In Table 4, we evaluate our model on the OPA dataset. As demonstrated in the table, our approach achieves the highest performance across all evaluated metrics.

**MagicBrush Results**. We evaluate our model on the MagicBrush test subset, which includes source and target prompts in addition to instructions. This allows us to compare our performance not only with instruction-following models like Hive and IP2P but also with prompt-based models like VQGAN-CLIP and SDEdit. As presented in Table 2, our model achieves the best results in most target image similarity metrics ($L_1$, CLIP-I, DINO and CMMD). The target prompts also allow us to compare the CLIP-T metric. While our model surpasses most methods in this metric, VQGAN-CLIP significantly outperforms it. This result is expected as the latter maximizes an equivalent objective during the editing process. Although some methods outperform ours in CLIP-T, they fall behind in other metrics. To highlight our model's superior balance between consistency with the original image and following the instruction, we present comparisons in fig. 6. As shown, our method outperforms all others in this tradeoff. Following (Zhang et al., 2024), we also fine-tuned our model on the object-addition training subset of MagicBrush and compared it against the similarly fine-tuned IP2P, with our model exceeding IP2P in all metrics.

Evaluations across the benchmarks show our model consistently outperforms competitors, affirming not only its high-quality outputs but also its robustness and adaptability across varied domains.

| Methods | L1$_\downarrow$ | L2$_\downarrow$ | CLIP-I$_\uparrow$ | DINO$_\uparrow$ | CLIP-T$_\uparrow$ | CMMD$_\downarrow$ |
|---|---|---|---|---|---|---|
| VQGAN-CLIP Crowson et al. (2022) | .211 | .078 | .670 | .507 | **.484** | .862 |
| SDEdit Meng et al. (2021) | .168 | .057 | .765 | .572 | .325 | .539 |
| Null-Text-Inversion Mokady et al. (2023) | **.072** | **.017** | .877 | .817 | .299 | .303 |
| Pix2PixZero Parmar et al. (2023) | .086 | .024 | .846 | .750 | .294 | .322 |
| EF-DDPM Huberman-Spiegelglas et al. (2024) | .110 | .030 | .844 | .716 | .328 | .342 |
| Hive Zhang et al. (2023) | .095 | .026 | .846 | .782 | .297 | .353 |
| IP2P Brooks et al. (2023) | .100 | .031 | .860 | .766 | .289 | .363 |
| Ours | **.072** | .025 | **.900** | **.852** | .302 | **.301** |
| *Fine-tune on MagicBrush* | | | | | | |
| IP2P Zhang et al. (2024) | .077 | .028 | .902 | .867 | .306 | .352 |
| Ours | **.067** | **.023** | **.910** | **.897** | **.308** | **.298** |

Table 2: **Results on MagicBrush** Top: Our model and various baselines tested on the MagicBrush test set subset. Bottom: Our model and IP2P fine-tuned on MagicBrush and tested on the subset.

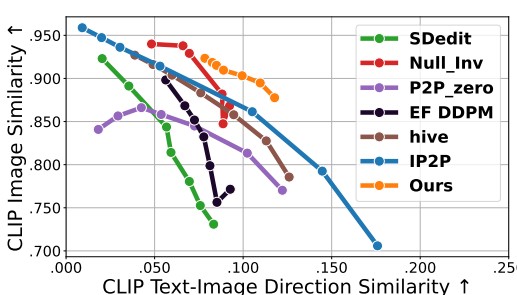

Figure 6: **Consistency-Instruction Trade-off on MagicBrush Subset.**

Table 3: **Results on PIPE Test Set.**

| Methods | L1$_\downarrow$ | L2$_\downarrow$ | CLIP-I$_\uparrow$ | DINO$_\uparrow$ | CMMD$_\uparrow$ |
|---|---|---|---|---|---|
| Hive | .088 | .021 | .849 | .754 | .232 |
| IP2P | .098 | .027 | .861 | .753 | .142 |
| Ours | **.057** | **.014** | **.945** | **.903** | **.060** |

Table 4: **Results on OPA.**

| Methods | L1$_\downarrow$ | L2$_\downarrow$ | CLIP-I$_\uparrow$ | DINO$_\uparrow$ | CMMD$_\uparrow$ |
|---|---|---|---|---|---|
| Hive | .126 | .041 | .802 | .670 | .481 |
| IP2P | .109 | .035 | .806 | .647 | .467 |
| Ours | **.084** | **.027** | **.848** | **.735** | **.360** |

### 5.3 Qualitative Examples

Fig. 3 qualitatively compares our model with other top-performing models across several datasets. The results illustrate how the proposed model, in contrast to competing approaches, seamlessly adds synthesized objects into images naturally and coherently, while maintaining consistency with the original images before editing. Furthermore, the examples, along with those in Figure 1, demonstrate our model's ability to generalize beyond its training classes, successfully integrating items such as a "princess" and "buttoned shirt". Additional examples are provided in the appendix.

### 5.4 Qualitative Evaluation

To complement the quantitative analysis, we conduct a human evaluation survey, comparing our model to IP2P. To this end, we randomly sample 100 images from the Conceptual Captions dataset (Sharma et al., 2018) and request human annotators to provide reasonable addition instructions. Next, we perform the edits using both models and request a different set of human evaluators to review their success. We adopt the queries from (Zhang et al., 2024) and ask evaluators to assess two aspects: alignment faithfulness between results and edit requests, and the output's general quality and consistency. Overall, we collected $1,833$ individual responses from $57$ different human evaluators, all participants from a pool of random internet users. To minimize biases and ensure an impartial evaluation, they completed the survey unaware of the research goals. We quantify edit faithfulness and output quality using two metrics: (i) overall global preference measured in percentage and (ii) aggregated per-image preference in absolute numbers (summed to $100$). The results in Table 5 showcase a substantial preference by human observers for our model's outputs in both following instructions and image quality. On average, the global preference metric indicates that our model is preferred approximately $72.6\%$ of the time. Additional survey details are provided in the supplementary materials. An additional human evaluation against hive is presented in table S8.

### 6 Leveraging PIPE for General Editing

We explore the application of our dataset in the broader context of image editing, extending its use beyond merely object addition. We combine the IP2P general editing dataset with PIPE and use it

Table 5: **Human Evaluation.** Comparison of our model with IP2P on edit faithfulness and quality. "Overall" represents the total vote percentage. "Per-image" quantifies the number of images where a model's outputs were preferred.

| Methods | Edit faithfulness | | Quality | |
|---|---|---|---|---|
| | Overall [%] | Per-image | Overall [%] | Per-image |
| IP2P | 26.4 | 28 | 28.5 | 31 |
| Ours | **73.6** | **72** | **71.5** | **69** |

Table 6: **General Editing Results on MagicBrush Test Set.** Model performance Evaluation on the Full General Editing MagicBrush test set. The model, trained on the combined PIPE and IP2P dataset and fine-tuned on the MagicBrush training set, surpasses the previously top-performing fine-tuned IP2P, demonstrating the potential of PIPE for enhancing general editing performance.

Figure 7: **General Editing Consistency-Instruction Trade-off.** Trade-off between consistency to input image (Y-axis) and edit adherence (X-axis), with text guidance fixed at 7 and varying image guidance $[1, 2.5]$.

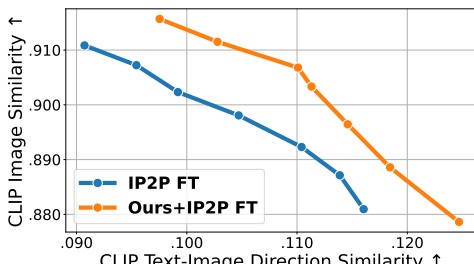

| Methods | L1$_\downarrow$ | L2$_\downarrow$ | CLIP-I$_\uparrow$ | DINO$_\uparrow$ | CLIP-T$_\uparrow$ |
|---|---|---|---|---|---|
| IP2P | .112 | .037 | .842 | .745 | .291 |
| IP2P FT | .082 | .032 | .896 | .845 | .301 |
| Ours+IP2P FT | **.074** | **.026** | **.906** | **.866** | **.303** |

to train an editing diffusion model, following the procedure outlined in Section 4. For evaluation, we utilized the entire MagicBrush test set, comparing our model against the IP2P model, both with and without MagicBrush fine-tuning. Diverging from the object addition concentrated approach, the model is fine-tuned using the full MagicBrush training set. To ensure fairness and reproducibility, all models were run with the same seed. Evaluations were conducted using the script provided by (Zhang et al., 2024), and the official models were employed with their recommended inference parameters. As illustrated in Table 6, our model sets new state-of-the-art scores for the general editing task, surpassing the current leading models. As presented in Figure 7, our fine-tuned model surpasses the current leading IP2P fine-tuned model, demonstrating higher image consistency for the same directional similarity values. The results collectively affirm that the PIPE dataset can be combined with any editing dataset and improve overall performance. In the appendix, we provide a qualitative visual comparison, showcasing the enhanced capabilities of the new model, not limited to object addition, as well as similar plots for the object addition subset used in Section 5.

## 7 LIMITATIONS

Despite the impressive results produced by our model, several limitations remain. First, while our data curation pipeline improves robustness during the removal phase, it is not entirely error-free. Additionally, the model struggles with significant changes occurring far from the object but are affected by it. For instance, it handles nearby effects, like TV shadows (see fig. 1 and fig. S14), but struggles with larger shadows or distant reflections, as seen in the center images of fig. S14. Similarly, object-object interactions are not always accurately handled (see the right images in the figure). These challenges stem from the dataset construction, as our method minimizes alterations outside the near-object region. Future work could explore inpainting both the object and distant regions influenced by it. We hope our work inspires future research to address these limitations.

## 8 DISCUSSION

In this work, we introduce the Paint by Inpaint framework, which identifies and leverages the fact that adding objects to images is fundamentally the inverse process of removing them. Building on this insight, by harnessing the wealth of available segmentation datasets and utilizing a high-performance mask-based inpainting model, we present PIPE, an object addition dataset. Unlike other mask-free, instruction-following editing datasets, PIPE is both large-scale and features consistent and natural editing target images. We demonstrate that training a diffusion model on the dataset leads to state-of-the-art performance in instruction-based image editing, proving the value of the PIPE dataset in achieving consistent and realistic image edits.

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

APPENDIX

## A ADDITIONAL MODEL OUTPUTS

In continuation of the demonstrations seen in Figure 1, we further show a variety of object additions performed by our model in Figure S8. The editing results showcase the model's ability to not only add a diverse assortment of objects and object types but also to integrate them seamlessly into images, ensuring the images remain natural and appealing.

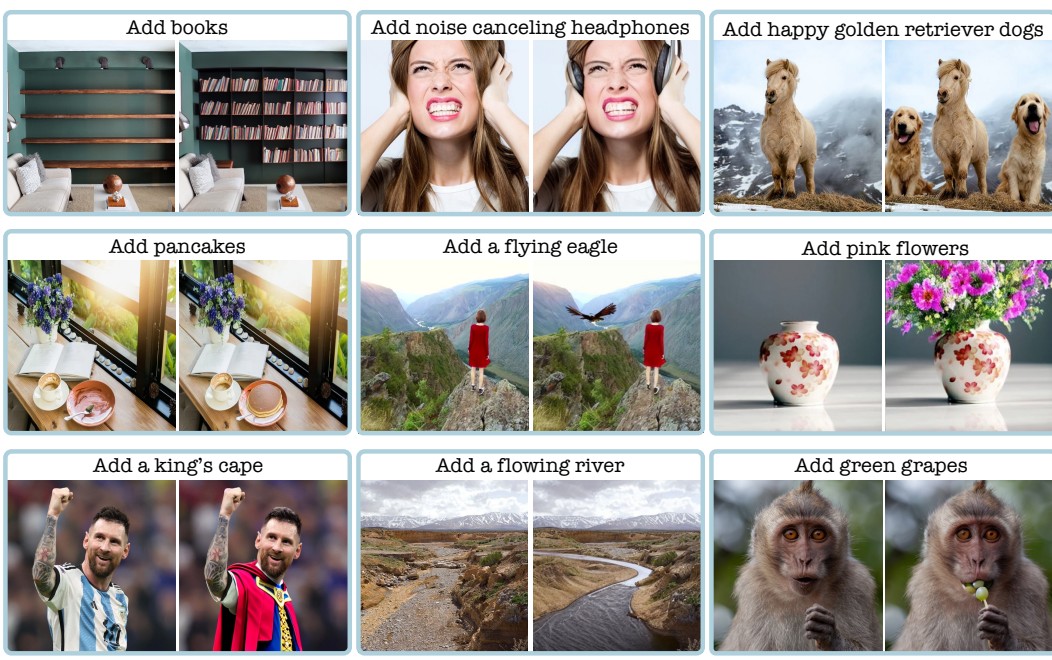

Figure S8: **Additional Object Addition Results of the Proposed Model.** The first two rows showcase outcomes from the model trained only with the PIPE dataset. The last row presents results from the same model after fine-tuning on the MagicBrush training set, as detailed in Section 5.2.

## B PIPE DATASET

### B.1 CREATING SOURCE-TARGET IMAGE PAIRS

We offer additional details on the post-removal steps described in Section 3.1. The post-removal process involves assessing the CLIP similarity between the class name of the removed object and the inpainted area. This assessment helps evaluate the quality of the object removal, ensuring no objects from the same class remain. To measure CLIP similarity for the inpainted area only, we counter the challenge of CLIP's unfamiliarity with masked images by reducing the background's influence on the analysis. We do this by adjusting the background to match the image's average color and integrating the masked area with this unified background color. A dilated mask smoothed with a Gaussian blur is employed to soften the edges, facilitating a more seamless and natural-looking blend.

To complement the CLIP score similarity, we introduce an additional measure that quantifies the shift in similarity before and after removal. Removals with a high pre-removal similarity score, followed by a comparatively lower yet significant post-removal score are not filtered, even though they exceed the threshold. This method allows for the efficient exclusion of removals, even when other objects of the same class are in close spatial proximity.

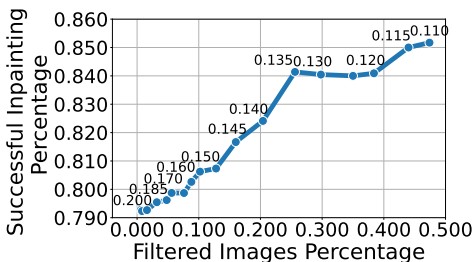

Figure S9: **Pre-Removal Filtered Examples.** Left: Objects with non-informative view and low CLIP Object similarity. Right: Extremely small and large objects, unsuitable for our dataset.

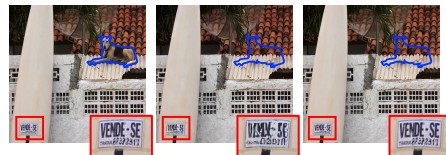
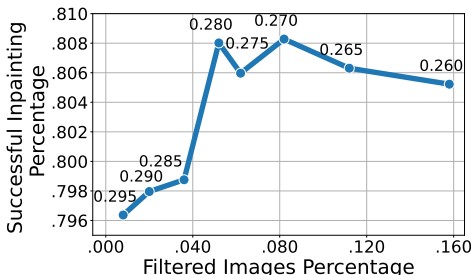

Figure S10: **Consistency Enforcement Examples.** From left to right: original image, inpainted dog image, inpainted image after alpha blending.

Figure S11: **Concensus Filtering Success for varying Thresholds**

Figure S12: **Multimodal CLIP Filtering Success for varying Thresholds**

### B.2 VLM-LLM BASED INSTRUCTIONS

Using a VLM and an LLM, we convert the class names of objects from the segmentation dataset into detailed natural language instructions (Section 3.2). Initially, for each image, we present the masked image (featuring only the object) to CogVLM with the prompt: `"Accurately describe the main characteristics of the <class name>. Use few words which best describe the <class- name>"`. This process yields an in-depth description centered on the object, highlighting key attributes such as shape, color, and texture. Subsequently, this description is provided to the LLM along with human-crafted prompts for In-Context Learning (ICL), to generate succinct and clear instructions. The implementation of the ICL mechanism is detailed in Table S7.

Furthermore, we enrich the instructions by including a coarse language-based description of the object's location within the image, derived from the given mask. To accomplish this, we split the image into a nine-section grid and assign each section a descriptive label (e.g., top-right). This spatial description is then randomly appended to the instruction with a 25% probability during the training process.

### B.3 INTEGRATING INSTRUCTION TYPES

As detailed in Section 3.2, we construct our instructions using three approaches: (i) class name-based (ii) VLM-LLM based, and (iii) manual reference-based. These three categories are then integrated to assemble the final dataset. The dataset includes 887,773 instances each from Class name-based and VLM-LLM-based methods, with an additional 104,373 from Manual reference-based instructions.

### B.4 ADDITIONAL EXAMPLES

In Figure S13, we provide further instances of the PIPE dataset that complement those in Figure 5.

## C IMPLEMENTATION DETAILS

As noted in Section 4, the training of our editing model is initialized with the SD v1.5 model. Conditions are set with $c_T = \varnothing$, $c_I = \varnothing$, and both $c_T = c_I = \varnothing$ occurring with a 5% probability

Table S7: **In-Context Learning Prompt**. (Top) We provide the model with five examples of captions and their corresponding human-annotated responses. (Bottom) We introduce it with a new caption and request it to provide an instruction.

---

[**USER**]: Convert the following sentence into a short image addition instruction:
¡caption 0¿.
Use straightforward language and describe only the ¡class name 0¿.
Ignore surroundings and background and avoid pictorial description.
[**ASSISTANT**]: ¡example response 0¿

$\vdots$

[**USER**]: Convert the following sentence into a short image addition instruction:
¡caption 4¿.
Use straightforward language and describe only the ¡class name 4¿.
Ignore surroundings and background and avoid pictorial description.
[**ASSISTANT**]: ¡example response 4¿

- - - - - - - - - - - - - - - - - - - - - - - - - - - - - - - - - - - - - - -

[**USER**]: Convert the following sentence into a short image addition instruction:
¡new caption¿.
Use straightforward language and describe only the ¡new class name¿.
Ignore surroundings and background and avoid pictorial description.
[**ASSISTANT**]:

---

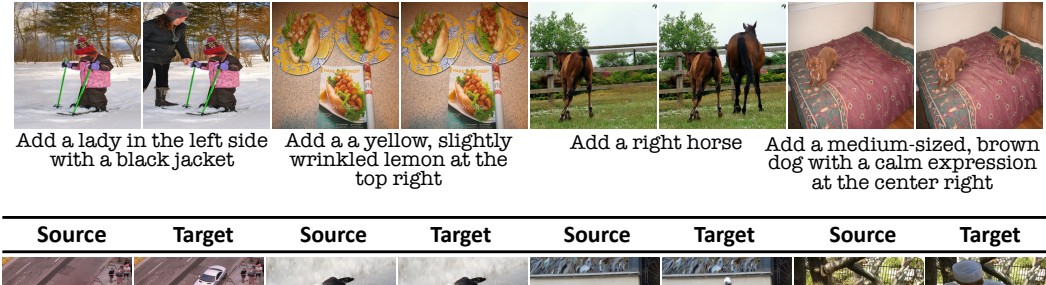

Figure S13: Additional **PIPE Datasets Examples.**

each. The input resolution during training is adjusted to 256, applying random cropping for variation. Each GPU manages a batch size of 128. The model undergoes training for 60 epochs, utilizing the ADAM optimizer. It employs a learning rate of $5 \cdot 10^{-5}$, without a warm-up phase. Gradient accumulation is set to occur over four steps preceding each update, and the maximum gradient norm is clipped at 1. Utilizing eight NVIDIA A100 GPUs, the total effective batch size, considering the per-GPU batch size, the number of GPUs, and gradient accumulation steps, reaches $4096 \, (128 \cdot 8 \cdot 4)$.

For the fine-tuning phase on the MagicBrush training set (Section 5.2), we adjust the learning rate to $10^{-6}$ and set the batch size to 8 per GPU, omitting gradient accumulation, and train for 250 epochs.

## C.1 MAGICBRUSH SUBSET

To initially focus our analysis on the specific task of object addition, we applied an automated filtering process to the MagicBrush dataset. This process aims to isolate image pairs and associated instructions that exclusively pertained to object addition. To ensure an unbiased methodology, we applied an automatic filtering rule across the entire dataset. The filtering criterion applied retained instructions explicitly containing the verbs "add" or "put," indicating object addition. Concurrently,

Figure S14: **Limitations.** Left: Successful shadow generation near the object. Center: Failures in generating shadows or reflections when distant from the object. Right: Failure in changing hand posture and maintaining the original one.

instructions with "remove" were excluded to avoid object replacement scenarios, and those with the conjunction "and" were omitted to prevent cases involving multiple instructions.

## C.2  EVALUATION

In our comparative analysis in Section 5.2, we assess our model against leading instruction-following image editing models. To ensure a fair and consistent evaluation across all models, we employed a fixed seed (0) for all comparisons.

Our primary analysis focuses on two instruction-guided models, IP2P (Brooks et al., 2023) and Hive (Zhang et al., 2023). For IP2P, we utilized the Hugging Face diffusers model and pipeline[4], adhering to the default inference parameters. Similarly, for Hive, we employed the official implementation provided by the authors[5], with the documented default parameters.

Our comparison extends to models that utilize global descriptions: VQGAN-CLIP (Crowson et al., 2022) Null-Text-Inversion (Mokady et al., 2023), Pix2PixZero (Parmar et al., 2023), Edit-Freindly DDPM (Huberman-Spiegelglas et al., 2024) and SDEdit (Meng et al., 2021). These models were chosen for evaluation within the MagicBrush dataset, as global descriptions are not available in both the OPA and our PIPE dataset. For VQGAN-CLIP[6], Null-Text-Inversion[7] and Edit-Freindly DDPM[8], we used the official code base with the default hyperparameters. For SDEdit[9] and Pix2PixZero[10], we used the image-to-image pipeline of the Diffusers library with the default parameters.

We also evaluated our fine-tuned model against the MagicBrush fine-tuned model, as documented in (Zhang et al., 2024). Although this model does not serve as a measure of generalizability, it provides a valuable benchmark within the specific context of the MagicBrush dataset. For this comparison, we employed the model checkpoint and parameters as recommended on the official GitHub repository of the MagicBrush project[11]. In Figure S15 and Figure S16, we provide additional qualitative examples on the tested datasets to complement the ones in Figure 3. We further assess the model's performance on the MagicBrush subset using the same CLIP Image similarity versus Directional CLIP similarity measure, as explained in Section 6. We plot this measure to compare the IP2P model with our model in Figure S17 and the MagicBrush fine-tuned models in Figure S18. As shown in both comparisons, our models present a better trade-off between consistency with the input image and adherence to the edit instruction, achieving higher consistency with the instruction for the same similarity to the input image.

---

[4]https://huggingface.co/docs/diffusers/training/instructpix2pix

[5]https://github.com/salesforce/HIVE

[6]https://github.com/nerdyrodent/VQGAN-CLIP

[7]https://github.com/google/prompt-to-prompt/blob/main/null_text_w_ptp.ipynb

[8]https://github.com/inbarhub/DDPM_inversion

[9]https://huggingface.co/docs/diffusers/en/api/pipelines/stable_diffusion/img2img

[10]https://huggingface.co/docs/diffusers/main/en/api/pipelines/pix2pix_zero

[11]https://github.com/OSU-NLP-Group/MagicBrush

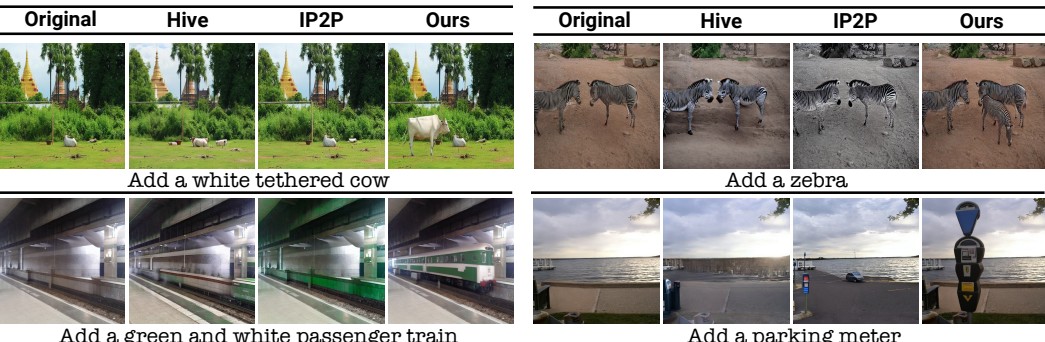

Figure S15: **Visual Comparison of the Proposed Model on PIPE Test Set.** The visual evaluation highlights the effectiveness of our method against other leading models on the PIPE test set. Our model excels in adhering closely to specified instructions and accurately generating objects in terms such as style, scale, and location.

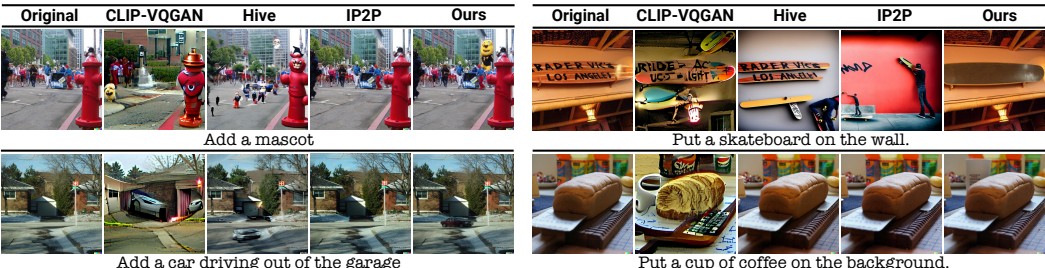

Figure S16: **Visual Comparison of the Proposed Model on MagicBrush Test Subset.** Our method versus leading models within the MagicBrush object addition test subset. It illustrates our model's superior generalization across varied instructions and datasets, outperforming the other approaches.

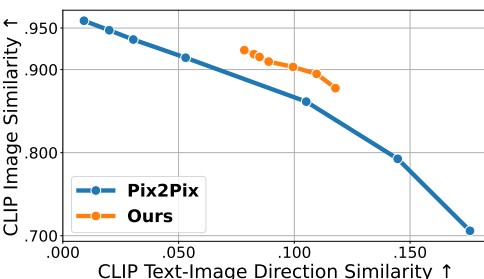

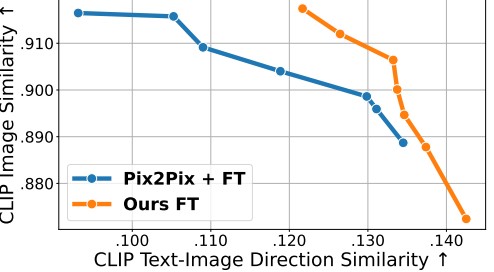

Figure S17: **Model Consistency-Instruction Trade-off:** Trade-off between consistency with the input image (Y-axis) and edit adherence (X-axis) for IP2P and our model on the MagicBrush test subset. Text guidance is fixed at 7, and image guidance ranges from 1 to 2.5.

Figure S18: **Finetuned-Model Consistency-Instruction Trade-off:** Trade-off between consistency with the input image (Y-axis) and edit adherence (X-axis) for IP2P and our model, both fine-tuned on the MagicBrush training set and tested on its test subset. Text guidance is fixed at 7, and image guidance ranges from 1 to 2.5.

## D  HUMAN EVALUATION

While quantitative metrics are important for evaluating image editing performance, they do not fully capture human satisfaction with the edited outcomes. To this end, we conduct a human evaluation survey, as explained in Section 5.4, comparing our model with IP2P and hive (table S8). Following (Zhang et al., 2024), we pose two questions: one regarding the execution of the requested edit and another concerning the overall quality of the resulting images. Figure S19 illustrates examples from our human survey along with the questions posed. Overall, our method leads to better results for human perception. Interestingly, as expected due to how PIPE was constructed, our model maintains a higher level of consistency with the original images in both its success and failure cases. For example, in the third row of Figure S19, while IP2P generates a more reliable paraglide, it fails to preserve the original background.

| Methods | Edit faithfulness | | Quality | |
| --- | --- | --- | --- | --- |
| | Overall [%] | Per image | Overall [%] | Per- image |
| Hive | 25.9 | 21 | 24.8 | 22 |
| Ours | **74.1** | **79** | **75.2** | **78** |

Table S8: **Human Evaluation against Hive.**

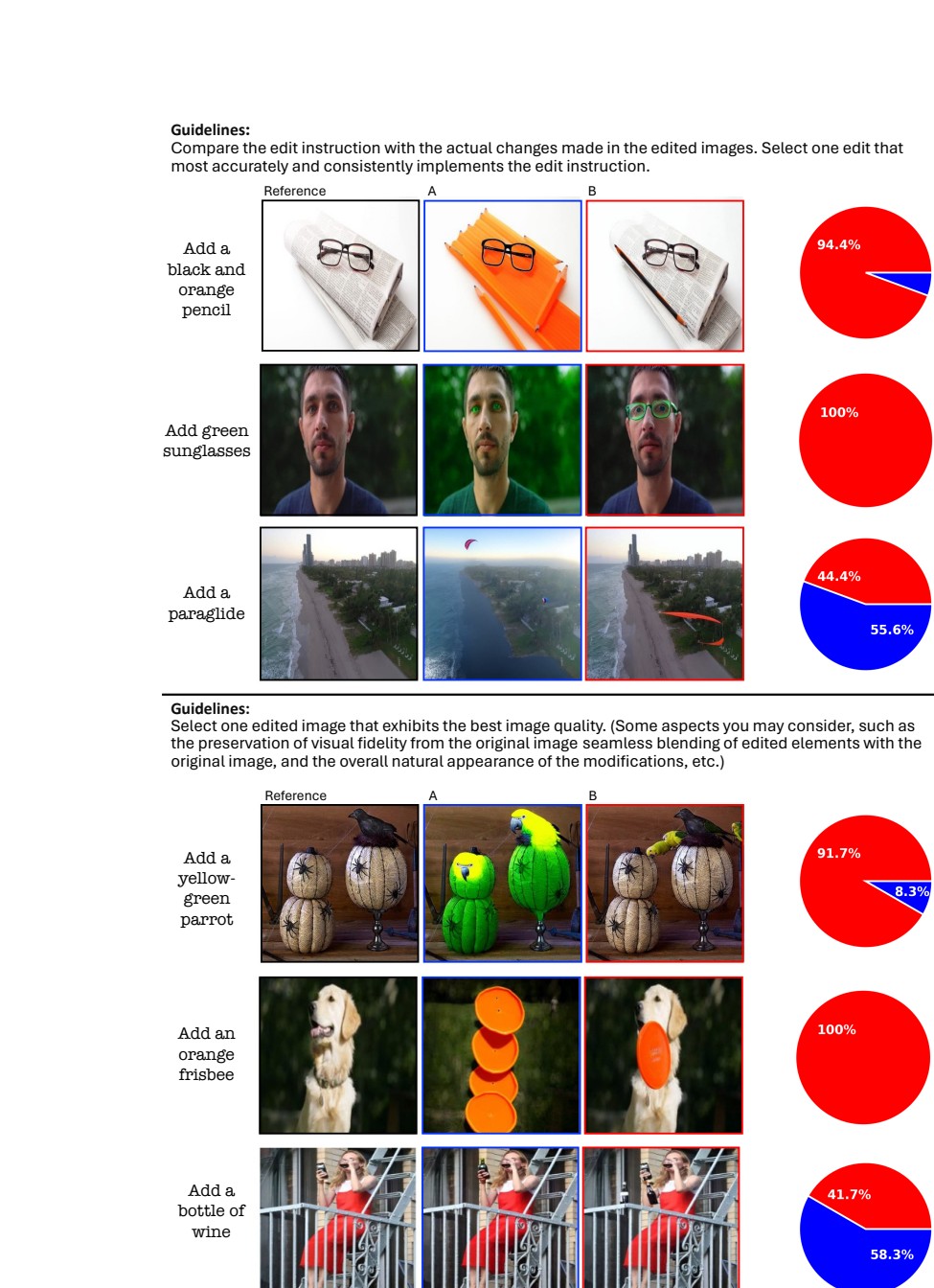

Figure S19: **Human Evaluation Examples**. Examples of the qualitative survey against IP2P alongside the response distribution (our method in red and the baseline in blue). The examples include both successful and failed cases of our model. The first three top examples correspond with a question focused on the edit completion, and the three bottom ones on the resulting image quality.

# E  INSTRUCTIONS ABLATION

We examine the impact of employing our VLM-LLM pipeline, detailed in Section 3.2, for generating natural language instructions. The outcomes of the pipeline, termed "long instructions", are compared with brief, class name-based instructions (*e.g.*, "Add a cat"), referred to as "short instructions". In Table S9, we assess a model trained on the PIPE image pairs, comparing its performance when trained with either long or short inputs. The models are evaluated on MagicBrush subset. As expected, training with long instructions leads to improved performance on MagicBrush. This demonstrates that training with comprehensive instructions generated by our VLM-LLM mechanism benefits at inference time. In addition to quantitative results, we provide qualitative results of both models in Figure S20. As illustrated, the model trained with long instructions shows superior performance in interpreting complex instructions that include detailed descriptions and location references, such as "Let's add a black bear to the stream".

| Original | Short Instructions | Long Instructions | Original | Short Instructions | Long Instructions |
|---|---|---|---|---|---|

Let's add a black bear to the stream.

Add red Mercedes-Benz bus with a large front windshield and an extended rear section

Figure S20: **Instructions Ablation Examples.** Qualitative comparison of model performance when trained on 'short' template-based instructions versus 'long' instructions generated through our VLM-LLM pipeline. Models trained on the latter exhibit superior performance in interpreting complex instructions and closely aligning object additions with editing requests.

| Train Instructions Type | L1 ↓ | L2 ↓ | CLIP-I ↑ | DINO↑ | CLIP-T ↑ |
|---|---|---|---|---|---|
| Short Instructions | 0.083 | 0.028 | **0.900** | **0.856** | 0.300 |
| Long Instructions | **0.072** | **0.025** | **0.900** | 0.852 | **0.302** |

Table S9: **Instructions Ablation Analysis.** A quantitative comparative analysis of model performance, comparing training on 'short' class-based instructions to 'long' instructions generated using the VLM and LLM pipeline. This analysis was performed on MagicBrush subset. The results demonstrate that training with VLM-LLM-based instructions significantly enhances performance, thereby confirming its effectiveness.

# F    GENERAL EDITING

As detailed in Section 6, the model, trained on the combined IP2P and PIPE dataset, achieves new state-of-the-art scores for the general editing task. In Figure S21, we present a visual comparison that contrasts our model's performance with that of a model trained without the PIPE dataset. The results not only underscore our model's superiority in object additions but also demonstrate its effectiveness in enhancing outcomes for other complex tasks, such as object replacement.

We further analyze this model by testing its performance not on the entire MagicBrush dataset as in Section 6, but on the 'addition only' subset (discussed in Appendix C.1) and its complementary 'not addition' subset. The experiments are performed under the same configuration as Section 6. Results for the addition subset and the complementary subset are presented in Table S10. In both subsets, our model outperforms the other models, indicating that although our dataset focuses on adding instructions, the inclusion of a large amount of high-quality editing data enhances performance for general editing tasks as well.

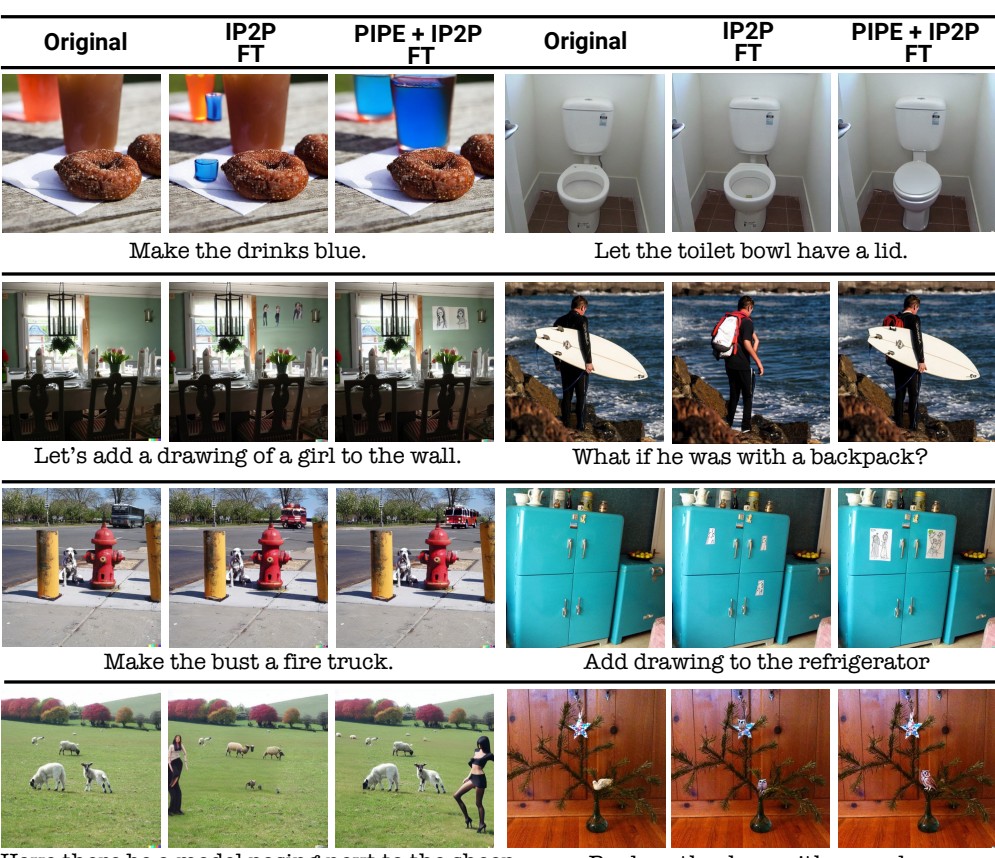

Figure S21: **Visual Comparison on General Editing Tasks.** The contribution of the PIPE dataset when combined with the IP2P dataset for general editing tasks, as evaluated on the full MagicBrush test set. The comparison is between a model trained on these merged datasets and a model trained solely on the IP2P dataset, with both models fine-tuned on the MagicBrush training set. The results demonstrate that, although the PIPE dataset focuses solely on object addition instructions, it enhances performance across a variety of editing tasks.

| | Addition Subset | | | | | Non-Addition Subset | | | | |
|---|---|---|---|---|---|---|---|---|---|---|
| Methods | L1$_\downarrow$ | L2$_\downarrow$ | CLIP-I$_\uparrow$ | DINO$_\uparrow$ | CLIP-T$_\uparrow$ | L1$_\downarrow$ | L2$_\downarrow$ | CLIP-I$_\uparrow$ | DINO$_\uparrow$ | CLIP-T$_\uparrow$ |
| IP2P | .100 | .031 | .860 | .700 | .289 | .114 | .038 | .839 | .742 | .290 |
| IP2P FT | .077 | .028 | .902 | .867 | .306 | .083 | .032 | .895 | .841 | .300 |
| Ours + IP2P FT | **.069** | **.024** | **.913** | **.889** | **.308** | **.075** | **.027** | **.905** | **.862** | **.303** |

Table S10: **Global Editing Performance on Addition and Non-Addition MagicBrush Subsets.** Evaluation of our global editing model performance on both the add and complementary non-add instruction subsets of MagicBrush. The model, trained on the combined PIPE and IP2P datasets and fine-tuned on the MagicBrush training set, surpasses IP2P and the fine-tuned IP2P models in both subsets.

## G  SOCIAL IMPACT AND ETHICAL CONSIDERATION

Using PIPE or the model trained with it significantly enhances the ability to add objects to images based on textual instructions. This offers considerable benefits, enabling users to seamlessly and quickly incorporate objects into images, thereby eliminating the need for specialized skills or expensive tools. The field of image editing, specifically the addition of objects, presents potential risks. It could be exploited by malicious individuals to create deceptive or harmful imagery, thus facilitating misinformation or adverse effects. Users are, therefore, encouraged to use our findings responsibly and ethically, ensuring that their applications are secure and constructive. Furthermore, PIPE, was developed using a VLM (Wang et al., 2023b) and an LLM (Jiang et al., 2023), with the model training starting from a SD checkpoint (Rombach et al., 2022). Given that the models were trained on potentially biased or explicit, unfiltered data, the resulting dataset may reflect these original biases.

