# OpenReview forum: "Paint by Inpaint: Learning to Add Image Objects by Removing Them First"
_ICLR.cc/2025/Conference — ICLR 2025 Conference Withdrawn Submission_

### Official Review · Reviewer_rxJK · 2024-10-31

**Soundness:** 2
**Presentation:** 3
**Contribution:** 2
**Rating:** 5
**Confidence:** 4

**Summary:**

The paper addresses the task of instruction-based image editing, with a specific focus on instructions that add new objects to the image.

To tackle this task, the authors note that it is much easier to remove an object from an image (e.g., using inpainting) than it is to add one, and so they can create a dataset of paired images with and without an object by using existing models to remove objects. To create editing instructions that align with the paired data, the authors leverage a VLM and an LLM. They further propose a set of quality filtering steps to ensure their dataset contains higher-quality examples. This dataset is used to train an instruction-based model (based on InstructPix2Pix) which outperforms existing baselines on object-adding benchmarks across a wide range of metrics.

Finally, the authors also demonstrate that their model can be further fine-tuned on general editing data (from e.g., MagicBrush) leading it to outperform InstructPix2Pix on general editing tasks.

**Strengths:**

The paper’s approach is reasonable, and it appears to outperform a diverse set of baselines on multiple datasets and across many metrics. In this sense, the comparisons shown in the paper are fairly comprehensive.

There is also significant detail provided in the paper and the supplementary which should help facilitate reproducibility, and the authors also promise to release their dataset which could help future research on this task.

Finally, the paper was relatively clear and easy to follow.

**Weaknesses:**

My primary concerns are twofold:

First, the core idea of learning to add objects by constructing synthetic data using inpainting already exists in prior published work, including work which was published in conference proceedings before the ICLR deadline (e.g., ObjectDrop, Winter et al, ECCV 2024). I think the authors should, at the very least, tone down their focus on presenting this as a core contribution, and instead highlight other aspects of the work such as the importance of the dataset and its construction.

Second, the authors repeatedly highlight advantages of their data construction approach, such as the fact that their target images are real and not synthetic, but I was unable to find any experiments which support these claims. Considering that the dataset and its construction are the paper’s core contribution, more should be done to ablate different aspects of its creation.

More specifically: the authors show that, as a whole, using their data during training is advantageous. They also ablate their use of VLM+LLM combo to create the editing instructions (in place of simple ‘add <class>’ type of prompts). However, I would have also liked to see experiments that demonstrate that: (1) The extensive data filtering the authors conduct is beneficial (at the very least, filtered data vs. unfiltered data, though if the authors have the compute for it, then more in-depth tests on specific filters would be interesting). (2) There is real advantage to using real images as targets, when compared with simply generating a larger set of synthetic examples.

The above concern is amplified by the fact that IP2P finetuned on MagicBrush data outperforms the proposed data construction method on most metrics, which signals that some aspects highlighted by the authors (e.g., fully realistic target images) are not as crucial.

Additional, more minor concerns:

a) I am unsure why the original IP2P is referred to as the state of the art on this task when MagicBrush (NeurIPS 2023) have a model (that the authors also use!) which outperforms it. The authors should add comparisons to this model in the qualitative figures, and I’m not sure I understand the reasoning behind arbitrarily splitting it to a different comparison in the quantitative results. The user study experiment should likely also be done against MagicBrush and not against the original IP2P.

b) The images in the paper show noticeable degradation of quality. The images lose a lot of high frequency detail and faces seem distorted (see e.g., lines in the carpet or the hair of Tobey Maguire in Fig 1.). Is this mostly a result of the VAE? Can you report fidelity metrics (e.g., L2) on the background for just VAE reconstruction?

Nits (did not affect my score but should be fixed):

a) In line 400: CLIP is not by Ruiz et al, and they are not remotely the first to use CLIP-T to evaluate image-text alignment, not even in the context of diffusion models or even in the context of model customization or image editing. This line should probably just cite Radford et al.

b) When you cite as “Name et al., (Name et al, 2020)” please use \citet{} to cite as “Name et al. (2020),”.

**Questions:**

Please see the questions in the weakness section.

The main thing I’d like to know is what parts of your dataset construction approach are actually important. In other words, I'd like to see an ablation on the main contributions proposed by the paper.

---

### Official Review · Reviewer_KFEh · 2024-11-01

**Soundness:** 2
**Presentation:** 2
**Contribution:** 2
**Rating:** 5
**Confidence:** 3

**Summary:**

This paper addresses the challenge of seamlessly adding objects to images based on textual instructions without requiring user-provided input masks. The authors leverage the observation that removing objects (Inpainting) is easier than adding them (Painting) due to the availability of segmentation mask datasets. They develop an automated pipeline to create a large-scale dataset of image pairs, where one image is the original and the other is the object-removed version. Using this dataset, they train a diffusion model to reverse the inpainting process, effectively adding objects to images.

**Strengths:**

1. This paper constructs a dataset comprising original images, images with certain objects removed, and corresponding editing instructions.
2. Utilizing this dataset, the paper trains a network specifically designed to add objects to images.
3. The article is well-written, with clear and precise explanations.

**Weaknesses:**

1. In some scenarios, this method may not be applicable. For instance, if there are three tables in the image and I want to place a cup on one specific table, this approach might not work effectively.
2. The comparison method (such as SDEdit and Null-text inversion) were not specifically proposed for the task of adding objects through editing. Comparing with them may not be appropriate, and more suitable comparison methods should be included, such as BrushNet and other mask-based editing methods.

**Questions:**

As shown in Weaknesses.

---

### Official Review · Reviewer_Ffa1 · 2024-11-02

**Soundness:** 2
**Presentation:** 3
**Contribution:** 2
**Rating:** 3
**Confidence:** 4

**Summary:**

The paper presents an approach for object addition in image editing by reversing the process of inpainting (object removal). To achieve this, this paper automatically synthesize a dataset (PIPE), and introduce various filtering rules to clean the dataset. Experiment results demonstrate that the proposed method outperforms baselines on the task of object addition. User study is also performed, demonstrating that the proposed methods can achieve better results in terms of instruction faithfulness and visual quality.

**Strengths:**

- This paper introduces a dataset which is automatically generated by Stable Diffusion. This data might be helpful for the community to evaluate image editing methods.

- The filtering process might be useful. Researchers can use similar filtering process to get cleaner data.

- This paper provides evaluations through both automatic metrics and user study, validating that the proposed method outperforms baselines on the object addition task.

**Weaknesses:**

- The application scenarios of this method is too limited, as it only supports object addition. In contrast, the baselines mentioned in the paper, such as MagicBrush[1], can address various editing operations, including object addition, object replacement, object removal, action changes, and more. I suggest that the authors continue developing their method so that it can support at least 3-4 image editing operations.

- Object addition without requiring user-provided input masks is not a particularly challenging task, as it can be achieved through various methods. For example, Grounding-DINO could be used to locate the approximate position for the target object. In addition, user-provided masks can offer more precise locations with small overhead. *I.e.*, compared to text instruction, it does not take much extra effort for users to mark the target area.

- Applying the paint-by-inpaint mechanism in diffusion models is an important contribution of this paper. However, similar mechanism has been widely used in GAN series papers. For instance, GANs often use the image reconstruction task to train models to achieve image editing capabilities, such as ManiGAN[2] or SIMSG[3]. During the image reconstruction process, we can consider that the model automatically completes the process of object removal (inpainting) and addition (painting).

- Since the dataset is generated automatically by the inpainting model, Stable Diffusion, it is possible that the editing model has only learned to reverse Stable Diffusion’s process rather than  editing capabilities. I would like to see experiments on domain generalization. For example, the model could be trained on the dataset constructed by Stable Diffusion, and then evaluated on data generated by other inpainting methods (GLIDE, Imagen) or real-world images.


[1] Zhang, Kai, et al. "Magicbrush: A manually annotated dataset for instruction-guided image editing." NeurIPS 2024.
[2] Li, Bowen, et al. "Manigan: Text-guided image manipulation." CVPR 2020.
[3] Dhamo, Helisa, et al. "Semantic image manipulation using scene graphs." CVPR 2020.

**Questions:**

- The proposed method only supports object addition, which means the editing scenarios are too limited. I highly suggest the authors to enable additional editing operations on their proposed method.

- Out-of-domain experiments are necessary to evaluate the model's generalization and robustness.

---

### Official Review · Reviewer_VcgQ · 2024-11-03

**Soundness:** 3
**Presentation:** 3
**Contribution:** 2
**Rating:** 3
**Confidence:** 4

**Summary:**

This paper presents a diffusion model for text-based image editing that seamlessly adds objects into images without user-provided masks. Using a curated dataset of images paired with object-removed versions and diverse natural language instructions, the model achieves superior performance in object addition and editing tasks. The dataset and trained models will be released to advance future research.

**Strengths:**

1. It introduced of the Paint by Inpaint framework for image editing
2. It constructed PIPE, a large-scale, high-quality, mask-free, textual instruction-guided object addition image dataset.
3. The combination of Paint by Inpaint framework and the PIPE dataset can enhance the performance of adding objects to images.

**Weaknesses:**

1. Motivation: Currently, there are a lot of methods where new objects are added with the guidance of external signals such as bounding boxes. In this way, many attributes of the objects can be controlled, such as the size and the position of the new object. Are there any unique benefits from the proposed method when compared to these existing methods?

2. It seems that the proposed method is not very practical. Many of the attributes of the generated new object cannot be specified, such as the size, position, etc. The model just randomly add the object to where it "believes" the new object should be. It would be better if authors can share more information / evaluation towards these aspects.

3. The major contribution of the paper could be the new proposed dataset. The technical novelty is relatively weak in the currently version.

**Questions:**

please see weaknesses

---

### Note · Authors · 2024-11-13

I have read and agree with the venue's withdrawal policy on behalf of myself and my co-authors.